# The Pivotal Role of Long Noncoding RNA RAB5IF in the Proliferation of Hepatocellular Carcinoma via LGR5 Mediated β-Catenin and c-Myc Signaling

**DOI:** 10.3390/biom9110718

**Published:** 2019-11-08

**Authors:** Ja Il Koo, Hyo-Jung Lee, Ji Hoon Jung, Eunji Im, Ju-Ha Kim, Nari Shin, Deok Yong Sim, Jisung Hwang, Sung-Hoon Kim

**Affiliations:** 1College of Korean Medicine, Kyung Hee University, Seoul 02447, Korea; freelink78@naver.com (J.I.K.); hyonice77@naver.com (H.-J.L.); johnsperfume@khu.ac.kr (J.H.J.); ji4137@naver.com (E.I.); simdy0821@naver.com (D.Y.S.); hjsung0103@naver.com (J.H.); 2Department of East-West Medical Science Graduate School of Kyung Hee University, Yongin 17104, Korea; fragendear@naver.com

**Keywords:** long non-coding RNA, RAB5IF, LGR5, hepatocellular carcinoma, apoptosis

## Abstract

In the current study, the function of long noncoding RNA (LncRNA) RAB5IF was elucidated in hepatocellular carcinoma (HCCs) in association with LGR5 related signaling. Here TCGA analysis revealed that LncRNA RAB5IF was overexpressed in HCC, and its overexpression level was significantly (*p* < 0.05) correlated with poor prognosis in patients with HCC. Furthermore, LncRNA RAB5IF depletion suppressed cell proliferation and colony formation, increased sub G1 population, cleavage of poly ADP-ribose polymerase (PARP) and cysteine aspartyl-specific protease (caspase 3) and attenuated the expression of procaspase 3, pro-PARP and B-cell lymphoma 2 (Bcl-2) in HepG2 and Hep3B cells. Furthermore, LncRNA RAB5IF depletion reduced the expression of LGR5 and its downstreams such as β-catenin and c-Myc in HepG2 and Hep3B cells. Notably, LGR5 depletion also attenuated the expression of pro-PARP, pro-caspase3, β-catenin and c-Myc in HepG2 and Hep3B cells. Conversely, LGR5 overexpression upregulated β-catenin and c-Myc in Alpha Mouse Liver 12 (AML-12) normal hepatocytes. Overall, these findings provide novel evidence that LncRNA RAB5IF promotes the growth of hepatocellular carcinoma cells via LGR5 mediated β-catenin and c-Myc signaling as a potent oncogenic target.

## 1. Introduction

Hepatocellular carcinoma (HCC), the second leading cause of cancer death worldwide, represents 80–90% of all major liver cancer [1,2]. Most patients with HCC develop metastasis due to rapid progression, potent invasion and chemo-resistance to current chemotherapy [3,4]. Thus, it is critical to identify biomarkers for early diagnosis and prognosis for reducing the death rate of patients with HCC [5]. Emerging evidence reveals that long noncoding RNAs (LncRNAs), a class of noncoding RNAs more than 200 nucleotides in length, are involved in cancer progression [6,7], since several cancer risk loci are easily transcribed into LncRNAs for tumorigenesis [7,8]. Leucine-rich repeat-containing G protein-coupled receptor 5 (LGR5), also known as GPR49, belongs to the G-protein-coupled receptor (GPCR) family of proteins [9,10] and is involved in tumor initiation, proliferation and invasion in several tumors including HCCs as a stem cell biomarker [11] by activation of the Wnt/β-catenin pathway in several cancers including liver, cervical and glioma cancers [12,13,14]. Furthermore, it is well documented that c-Myc, one of oncogenes, is involved in transcription, translation, protein stability, regulated by twenty-five microRNAs and eighteen long noncoding RNAs in several cancers [15]. LncRNA RAB5IF has NR_026562 (Accession number), C20orf24 (Gene symbol), Exon sense-overlapping (Classification of LncRNAs) and human chromosome 20 location. Interestingly, C20orf24, a gene symbol of RAB5IF, was involved in the progression of colorectal carcinoma cells [16]. Nevertheless, the underlying molecular mechanism of LncRNA RAB5IF in HCC progression is still unclear to date, though TCGA revealed that LncRNA RAB5IF was up-regulated in HCC and high levels of LncRNA RAB5IF positively indicated poor prognosis in patients with HCC. Thus, in the present study, molecular mechanisms of RAB5IF were for the first time elucidated in association with LGR5 mediated β-catenin and c-Myc signaling axis in HepG2 and Hep3B HCCs.

## 2. Materials and Methods

### 2.1. Cell Culture

Cell lines were obtained from the American Type Culture Collection (ATCC). These cells were cultured in Dulbecco’s Modified Eagle Medium (DMEM) supplemented with 10% fetal bovine serum FBS and 1% antibiotic (Welgene, South Korea).

### 2.2. TCGA Data Analysis

The Cancer Genome Atlas (TCGA) data of LncRNA in Hepatocellular carcinoma were analyzed using TCGA data portal based on the paper of Falcon and his colleagues [17]. Here, a comprehensive bioinformatics analysis was conducted to identify LncRNA profile in hepatocellular carcinoma, using the RNA sequencing datasets collected from hepatocellular carcinoma patients and deposited at TCGA database (https//cancergenome.nih.gov). The expression profiles of 373 hepatocellular carcinoma tissues and 50 normal tissues were obtained from TCGA data portal. Data processing steps include inverse log (base 2) transformation and quantile normalization of expression values. In this study, the profiles of 373 cases were included in the survival analysis. Statistical analysis was performed with R software version 3.2.1. R core team (http://www.R-project.org/)

### 2.3. RNA Interference

HepG2 and Hep3B cells were transfected with LncRNA RAB5IF small interfering RNA (siRNA) oligonucleotides (sense: 3′-GAAUAGCAGGGAAAGGCCAtt-5′ and antisense: 5′-UGGCCUUUCCCUGCUAUUCcc-3′ (Thermo Fisher Scientific, Inc., Waltham, MA, USA) and LGR5 siRNA oligonucleotides (sense: 3′-CGUCUUCACCUCCUACCUA(dtdt) and antisense: 5′-UAGGUAGGAGGUGAAGACG(dtdt) (Bioneer, Daejeon, Korea) using INTERFERin (Polyplus-transfection SA, Illkirch, France) transfection reagent according to the manufacturer’s protocol.

### 2.4. RT-qPCR Analysis

Total RNA was isolated from LncRNA RAB5IF depleted or intact HepG2 and Hep3B cells by using RNeasy mini kit (Qiagen) and reverse-transcribed using M-MLV reverse transcriptase (Promega, Madison, WI). Quantitative reverse transcription PCR (RT-qPCR) was conducted with the LightCycler TM instrument (Roche Applied Sciences, Indianapolis, IN, USA) using the following primers, LncRNA RAB5IF- forward: 5′-AGTCTCCGTCTGGAGTAAGGTG−3′; reverse- 5′-CCTGCTATTCCCAAGAACCCTC–3′ (Bioneer, Daejeon, Korea), LGR5 forward: 5′- CGTTGCAACACTGTCATGGC-3′; reverse- 5- AGGTCAGGTGAAGCGCTCG−3′ (Bioneer, Daejeon, Korea), hGAPDH-forward 5′-CCA CTC CTC CAC CTT TGA CA-3′; reverse-5′-ACC CTG TTG CTG TAG CCA −3′ (Bioneer, Daejeon, Korea).

### 2.5. Cell Viability Assay

The cell viability of LncRNA RAB5IF depletion was measured by 3-(4,5-dimethylthiazol-2-yl)-2,5-diphenyltetrazolium bromide (MTT) assay. HepG2 and Hep3B cells transfected by LncRNA RAB5IF siRNA plasmid were seeded onto 96-well culture plate for 24 h. The cells were incubated with MTT (1 mg/mL) (Sigma Chemical, St. Louis, MO, USA) for 2 h and then treated with MTT lysis solution overnight. Optical density (OD) was measured using a microplate reader (Molecular Devices Co., Silicon Valley, CA, USA) at 570 nm.

### 2.6. Colony Formation Assay

HepG2 and Hep3B cells were transfected by LncRNA RAB5IF siRNA plasmid were seeded onto 6-well plates at a density of 1000 cells per well. After a week, the cells were stained with crystal violet (0.1% in phosphate-buffered saline (PBS)).

### 2.7. Cell Cycle Analysis

HepG2 and Hep3B cells transfected by LncRNA RAB5IF siRNA plasmid were seeded onto a 6-well culture plate for 24 h. The cells were washed twice with cold PBS and fixed in 75% ethanol at −20 °C. The cells were incubated with RNase A (10 mg/mL) for 1 h at 37 °C and stained with propidium iodide (50 μg/mL) for 30 min at room temperature in the dark. The stained cells were analyzed for the DNA content by FACSCalibur (Becton Dickinson, Franklin Lakes, NJ, USA) using CellQuest Software version 5.2.1 (Becton Dickinson, Franklin Lakes, NJ, USA).

### 2.8. Western Blotting

HepG2 and Hep3B cells were lysed in a lysis buffer (50 mM Tris–HCl, pH 7.4, 150 mM NaCl, 1% Triton X-100, 0.1% SDS, 1 mM EDTA, 1 mM Na_3_VO_4_, 1 mM NaF and 1× protease inhibitor cocktail) on ice, and spun down at 14,000× *g* for 20 min at 4 °C. The supernatants were collected and quantified for protein concentration by using RC DC protein assay kit (Bio-Rad, Hercules, CA, USA), The protein samples were separated on 4–12% NuPAGE Bis–Tris gels (Novex, Carlsbad, CA, USA) and transferred to a Hybond ECL transfer membrane for detection with antibodies for poly (ADP-ribose) polymerase (PARP), Caspase-3, c-Myc, LGR5, β-catenin and Bcl-2 (Santa Cruz Biotechnologies, Santa Cruz, CA, USA), and β-actin (Sigma, St. Louis, MO, USA).

### 2.9. Rescue Assay

For rescue assay, HepG2 and Hep3B cells were transfected with LncRNA RAB5IF siRNA for 48 h and then transfected with Lentivirus (Lv)-LncRNA RAB5IF and Lv-con viruses for 24 h.

### 2.10. Statistical Analysis

For statistical analysis of the data, GraphPad Prism software (GraphPad Software, Version 5.0, San Diego, CA, USA) was used. All data were expressed as means ± standard deviation (SD). Student’s *t*-test was used for comparison of two groups. The statistically significant difference was set at *p*-values of <0.05 between untreated control and treated groups

## 3. Results

### 3.1. LncRNA RAB5IF is Overexpressed in HCCs and Patient Tissues

TCGA analysis revealed that LncRNA RAB5IF was overexpressed compared to normal control database. LncRNA RAB5IF was significantly overexpressed in HCC tissues compared to normal tissues control (Figure 1a) by TCGA analysis. Furthermore, survival rate was significantly increased in HCC patients with low expression of LncRNA RAB5IF compared to the patients with its high expression (Figure 1b). Consistently, as shown in Figure 1c, LncRNA RAB5IF was highly expressed in HepG2, Hep3B and Huh7 cells, while that was weakly expressed in MCF-7, A549 and HeLa cell lines.

### 3.2. Depletion of LncRNA RAB5IF Inhibits Proliferation and Colony Formation of HCCs

To confirm whether LncRNA RAB5IF depletion suppresses proliferation and colony formation in HCCs, MTT assay and colony formation assay were conducted in HepG2 and Hep3B cells using LncRNA RAB5IF siRNA transfection assay. As shown in Figure 2a, LncRNA RAB5IF expression was significantly decreased by 90% in HepG2 and Hep3B cells after LncRNA RAB5IF siRNA transfection. Knockdown of LncRNA RAB5IF expression significantly suppressed proliferation and colony formation of HepG2 and Hep3B cells compared to untreated control (Figure 2b,c).

### 3.3. Depletion of LncRNA RAB5IF Induces Apoptosis in HCCs

To confirm whether antiproliferative effect of LncRNA RAB5IF depletion is due to apoptosis, cell cycle analysis was performed in LncRNA RAB5IF depleted HepG2 and Hep3B cells. LncRNA RAB5IF depletion increased sub-G1 population in HepG2 and Hep3B cells (Figure 3a). Consistently, a cell apoptosis assay using Annexin-V/PI staining revealed that LncRNA RAB5IF depletion increased the early and late apoptosis to 35.32% and 15.07% in HepG2 cells and 25.86% and 14.19% in Hep3B cells, respectively, compared to siControl (Figure 3b). Likewise, LncRNA RAB5IF depletion increased the cleavage of PARP and caspase3 and attenuated the expression of pro-PARP and pro-caspase 3 and Bcl-2 in HepG2 and Hep3B cells (Figure 3c).

### 3.4. Depletion of LncRNA RAB5IF Attenuates the Expression of LGR5, β-Catenin and c-Myc in HCCs

To confirm whether the apoptotic effect of LncRNA RAB5IF depletion is associated to LGR5, β-catenin and c-Myc, Western blotting was performed in HepG2 and Hep3B cells. Furthermore, LGR5 was highly expressed in HepG2, Hep3B and Huh7 cells, while weakly expressed in siHa cervical cancer, PC3 prostate cancer, A549 lung cancer and MCF-7 breast cancer cell lines. (Figure 4a). As shown in Figure 4b, LncRNA RAB5IF depletion attenuated the expression of LGR5, β-catenin and c-Myc in HepG2 and Hep3B cells. To confirm the pivotal role of LncRNA RAB5IF, rescue assay was performed in the HepG2 and Hep3B cell transfected by si-LncRNA RAB5IF and Lv-RAB5IF plasmids. LncRNA RAB5IF depletion reduced the expression of LGR5, β-catenin and c-Myc, while overexpression of Lv-LncRNA RAB5IF reversed the ability of LncRNA RAB5IF depletion to attenuate the expression of LGR5, β-catenin, c-Myc and pro-PARP in HepG2 and Hep3B cells (Figure 4c).

### 3.5. LGR5 Depletion Suppresses the Expression of β-Catenin and c-Myc in HCCs

To confirm the apoptotic effect of LGR5 depletion, Western blotting was performed in HepG2 and Hep3B cells. LGR5 depletion significantly suppressed the expression of β-catenin and c-Myc in HepG2 and Hep3B cells (Figure 5a). Consistently, LGR5 depletion attenuated Pro-PARP and Pro-caspase3 and decreased protein expressions of Bcl-2 in HepG2 and Hep3B cells (Figure 5a). Conversely, LGR5 overexpression upregulated the expression of β-catenin and c-Myc in Alpha Mouse Liver 12 (AML-12) normal hepatocytes (Figure 5b).

## 4. Discussion

Accumulating evidence reveals that many LncRNAs play critical roles in the initiation and progression of HCC [18,19]. Therefore, LncRNAs are on the spotlight as biomarkers and therapeutic targets for cancer therapy [20,21,22].

TCGA revealed that LncRNA RAB5IF was highly expressed in HCCs cell lines, while it was weakly expressed in breast, lung and cervical cell lines. Furthermore, recent evidence suggests that, C20orf24, a gene symbol of LncRNA RAB5IF, was involved in the progression of colorectal carcinoma cells [16].

Thus, the novel functions of LncRNA RAB5IF were examined in HCCs. Interestingly, LncRNA RAB5IF was known as an uncharacterized gene to date and was overexpressed in neoplastic patient tissues including HCCs and also its overexpression level was significantly (*p* < 0.05) correlated with poor survival rate in patients with HCC by TCGA analysis.

Consistently, LncRNA RAB5IF was highly expressed in HepG2 and hep3B cells, while that was weakly expressed in MCF-7, A549 and HeLa cell lines, implying the oncogenic potential of LncRNA RAB5IF with poor survival rate in HCC patients.

It was also postulated that C20orf24, the gene symbol of LncRNA RAB5IF, may be most closely associated with LGR5, since they are derived from membrane proteins and also LGR5 was reported to be a stem cell biomarker and oncogenic molecule in HCCs [5,13,23]. Hence, to validate the oncogenic role of LncRNA RAB5IF, several experiments were conducted in HepG2, Hep3B and AML12 cell lines. Indeed, LncRNA RAB5IF depletion inhibited the proliferation, increased sub-G1 portion and attenuated pro-PARP and pro-caspase3 in HepG2 and Hep3B cells, demonstrating the antiproliferative and apoptotic effects of LncRNA RAB5IF depletion in HCCs.

LGR5, a member of the G protein-coupled receptor family of transmembrane receptors, is one of stem cell biomarkers especially in HCC. [23,24,25] Overexpression of LGR5 is implicated in proliferation, metastasis, epithelial-mesenchymal transition (EMT), cancer progression and poor prognosis of cancers [12,26,27]. Furthermore, LGR5 regulates Wnt/β-catenin and c-Myc as one of important target molecules in HCCs [13,27,28,29].

To verify that apoptosis induced by LncRNA RAB5IF depletion is mediated by LGR5 signaling, rescue assay was conducted in HepG2 and Hep3B cells. Here LncRNA RAB5IF depletion attenuated the expression of LGR5, its downstream oncogenic molecules such as β-catenin and c-Myc and pro-PARP in HepG2 and Hep3B cells, while overexpression of RAB5IF recovered the expression of β-catenin and c-Myc and pro-PARP suppressed by LncRNA RAB5IF depletion compared to Lv-Control, demonstrating that LncRNA RAB5IF depletion induced apoptosis can be mediated by inhibition of LGR5 mediated β-catenin and c-Myc signaling axis in HepG2 and Hep3B cells.

To prove the crucial role of LGR5, LGR5 overexpression plasmid was transfected into normal hepatocyte AML-12 cells. As expected, LGR5 overexpression reversed the ability of LncRNA RAB5IF depletion to attenuate the expression of β-catenin and c-Myc in AML-12 cells, implying that LGR5 mediates the apoptotic effect of LncRNA RAB5IF in HCCs.

However, given that several miRNAs regulate target proteins including LGR5 [30], it can be postulated that LncRNA RAB5IF may regulate LGR5 as a LGR5 targeted miRNA sponge, since LncRNA RAB5IF depletion reduced protein expression of LGR5 in HepG2 and Hep3B cells. Similarly, Jing et al. [31] reported that LncRNA CASC15 works as a sponge to inhibit miR-4310 that targets LGR5. However, further study is required to confirm the role of LGR5 targeted miRNA sponge by LncRNA RAB5IF using advanced molecular works including in silico analysis in the future. In summary, LncRNA RAB5IF was overexpressed in HCCs along with poor prognosis in patients with HCC by TCGA analysis. Consistently, LncRNA RAB5IF depletion suppressed cell proliferation and colony formation, increased sub G1 population and attenuated the expression of procaspase 3, pro-PARP and Bcl-2 in HepG2 and Hep3B cells. Notably, LncRNA RAB5IF depletion reduced the expression of LGR5, β-catenin and c-Myc in HepG2 and Hep3B cells. Furthermore, LGR5 depletion also attenuated pro-PARP, pro-caspase3 and Bcl-2 in HepG2 and Hep3B cells and LGR5 overexpression upregulated β-catenin and c-Myc in AML-12 cells. Taken together, these findings for the first time demonstrate novel evidence that LncRNA RAB5IF promotes the growth of hepatocellular carcinoma cells via upregulation of LGR5 mediated β-catenin and c-Myc signaling axis as a potent oncogenic target (Figure 6).

## 5. Conclusions

Our findings suggest that LncRNA RAB5IF was overexpressed in HCCs along with poor prognosis in patients with HCC by TCGA analysis. Consistently, LncRNA RAB5IF depletion suppressed cell proliferation and colony formation, increased sub G1 population and attenuated the expression of procaspase 3, pro-PARP and Bcl-2 in HepG2 and Hep3B cells. Notably, LncRNA RAB5IF depletion reduced the expression of LGR5, β-catenin and c-Myc in HepG2 and Hep3B cells. Furthermore, LGR5 depletion also attenuated pro-PARP, pro-caspase3 and Bcl-2 in HepG2 and Hep3B cells and LGR5 overexpression upregulated β-catenin and c-Myc in AML-12 cells. Taken together, these findings for the first time demonstrate novel evidence that LncRNA RAB5IF promotes the growth hepatocellular carcinoma cells via upregulation of LGR5 mediated β-catenin and c-Myc signaling axis as a potent oncogenic target.

## Figures and Tables

**Figure 1 biomolecules-09-00718-f001:**
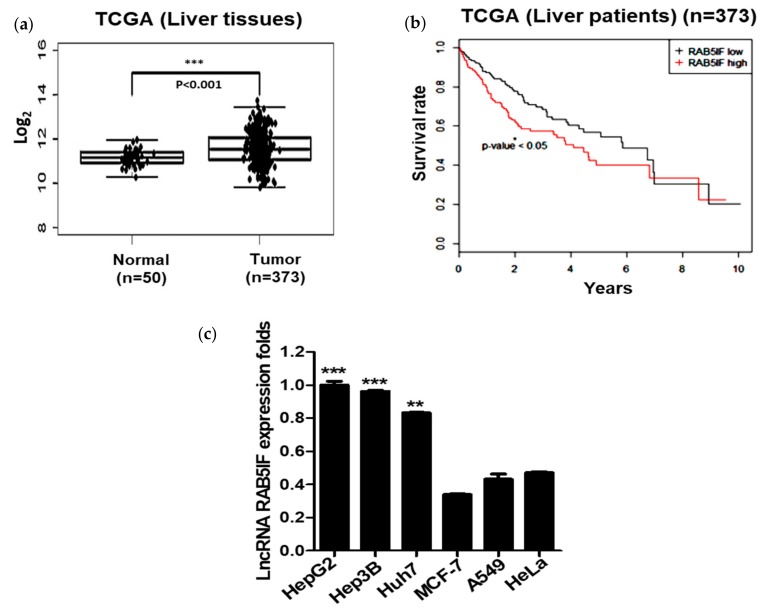
LncRNA RAB5IF is overexpressed in hepatocellular carcinoma patient tissues and hepatocellular carcinoma (HCC) cells. (**a**) The RNA expression of LncRNA RAB5IF in HCC tissues (*n* = 373) compared with normal tissues (*n* = 50) was overexpressed by The Cancer Genome Atlas (TCGA) analysis. Boxplots of log2-transformed (RPKM) gene expression values. Data represent means ± SD. *** *p* < 0.001. (**b**) Kaplan–Meier survival curve in tumor tissues (*n* = 373), as determined according to LncRNA RAB5IF expression level. Data represent means ± standard deviation (SD). * *p* < 0.05. (**c**) LncRNA RAB5IF expression levels in various human cancer cell lines by quantitative real time polymerase chain reaction (qRT-PCR). Data represent means ± SD by two independent experiments. ** *p* < 0.01 and *** *p* < 0.001 vs. LncRNA RAB5IF level in MCF-7 cells.

**Figure 2 biomolecules-09-00718-f002:**
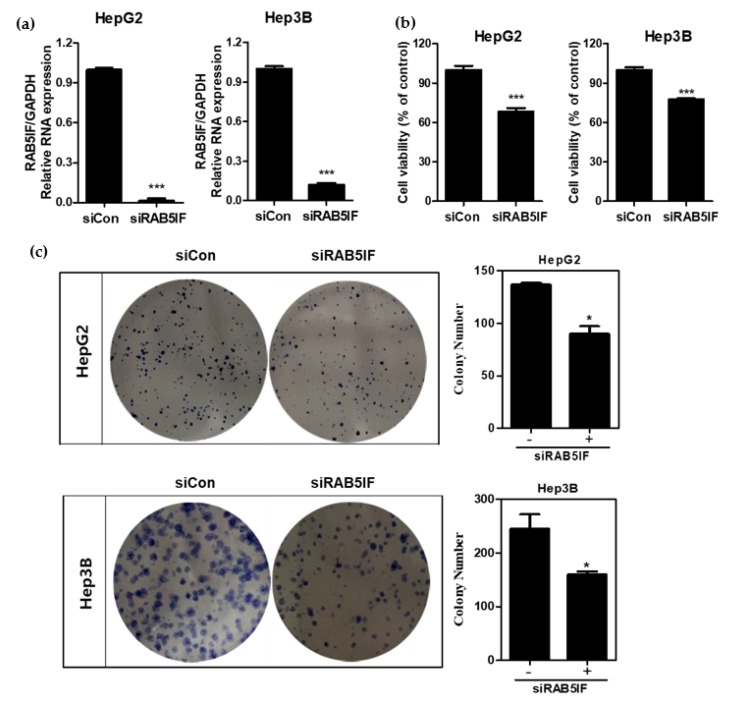
LncRNA RAB5IF depletion suppresses proliferation and colony formation in HCCs. (**a**) The efficiency of siRNA transfection targeting LncRNA RAB5IF in HepG2 and Hep3B cells was detected by qRT-PCR. Data represent means ± SD. (Two independent expreriments). *** *p* < 0.001. (**b**) Effect of LncRNA RAB5IF depletion on the cell viability of HepG2 and Hep3B cells by MTT assay. Data represent means ± SD by two independent experiments. *** *p* < 0.001 vs. siRNA control. (**c**) Photos for colony formation and bar graph (right) for colony formation in LncRNA RAB5IF depleted HepG2 and Hep3B cells. The colonies were visualized and counted by staining with crystal violet. Data represent means ± SD by two independent experiments. * *p* < 0.05 vs. siRNA control.

**Figure 3 biomolecules-09-00718-f003:**
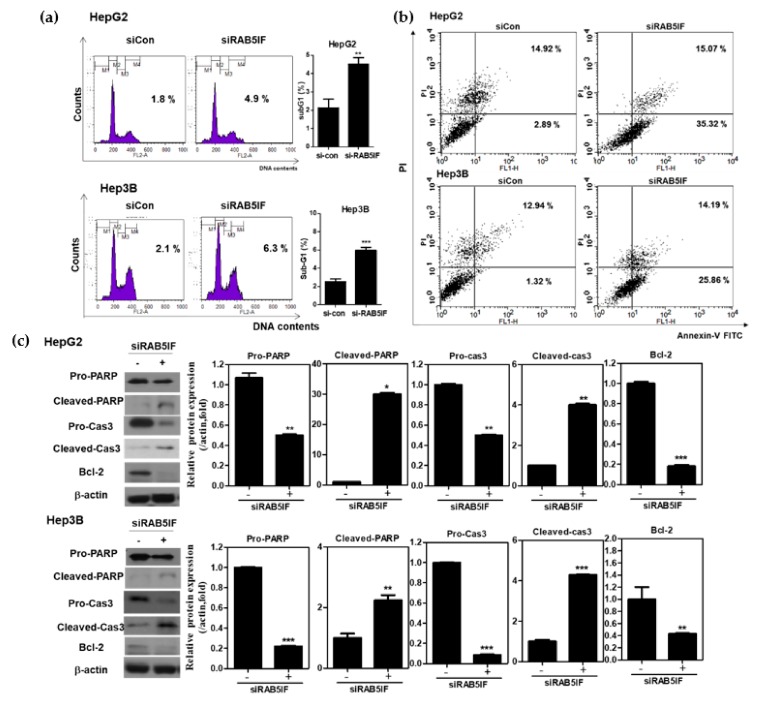
Depletion of LncRNA RAB5IF induces apoptosis in HCCs. (**a**) Effect of LncRNA RAB5IF depletion on cell cycle distribution in HepG2 and Hep3B cells by Fluorescence-activated cell sorting (FACS). Data represent means ± SD by three independent experiments. ** *p* < 0.01 and *** *p* < 0.001 vs. siRNA control. (**b**) After transient transfection of HepG2 and Hep3B cells with LncRNA RAB5IF siRNA, Annexin- Propidium Iodide (PI)staining assays was performed. The cells were stained using Fluorescein isothiocyanate (FITC)-Annexin V/PI dye and early and late apoptotic portions were detected by flow cytometry. (**c**) Effect of LncRNA RAB5IF depletion on apoptosis related genes in HepG2 and Hep3B cells by Western blotting. Cell lysates were prepared and subjected to Western blotting for PARP, Caspase3 and Bcl-2. Graphs represent relative level of protein expression /β-actin. Data represent means ± SD by two independent experiments. * *p* < 0.05, ** *p* < 0.01 and *** *p* < 0.001 vs. siRNA control.

**Figure 4 biomolecules-09-00718-f004:**
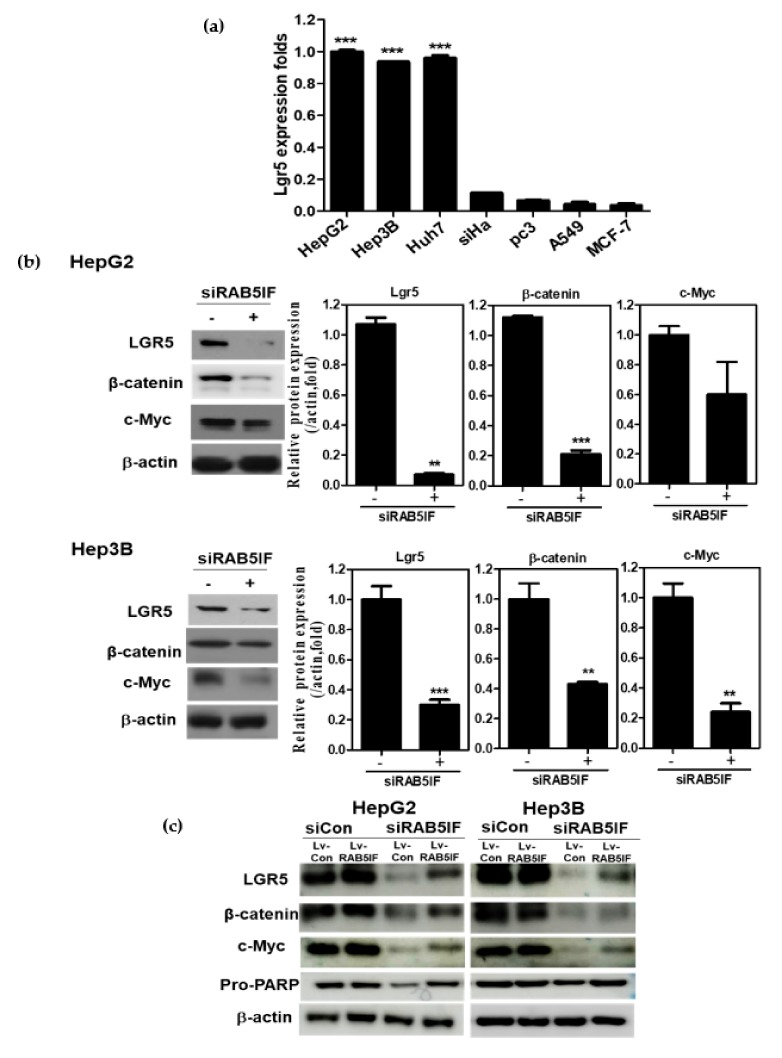
LncRNA RAB5IF regulates c-Myc and LGR5 in HCCs. (**a**) LGR5 RNA expression in various human cancer cell lines by qRT-PCR. Data represent means ± SD by two independent experiments. ** *p* < 0.01 and *** *p* < 0.001 vs. LGR5 level in MCF-7 cells. (**b**) Effect of LncRNA RAB5IF depletion on LGR5 pathway in HepG2 and Hep3B cells by Western blotting. Cell lysates were prepared and subjected to Western blotting for LGR5, β-catenin and c-Myc. Graphs represent relative level of protein expression /β-actin. Data represent means ± SD by two independent experiments. ** *p* < 0.01 and *** *p* < 0.001 vs. siRNA control. (**c**) HepG2 and Hep3B cells were transfected with LncRNA RAB5IF siRNA for 48 h and then transfected with Lv-LncRNA RAB5IF and Lv-con viruses for 24 h. Cell lysates were prepared and subjected to Western blotting for LGR5, β-catenin, c-Myc and Pro-PARP.

**Figure 5 biomolecules-09-00718-f005:**
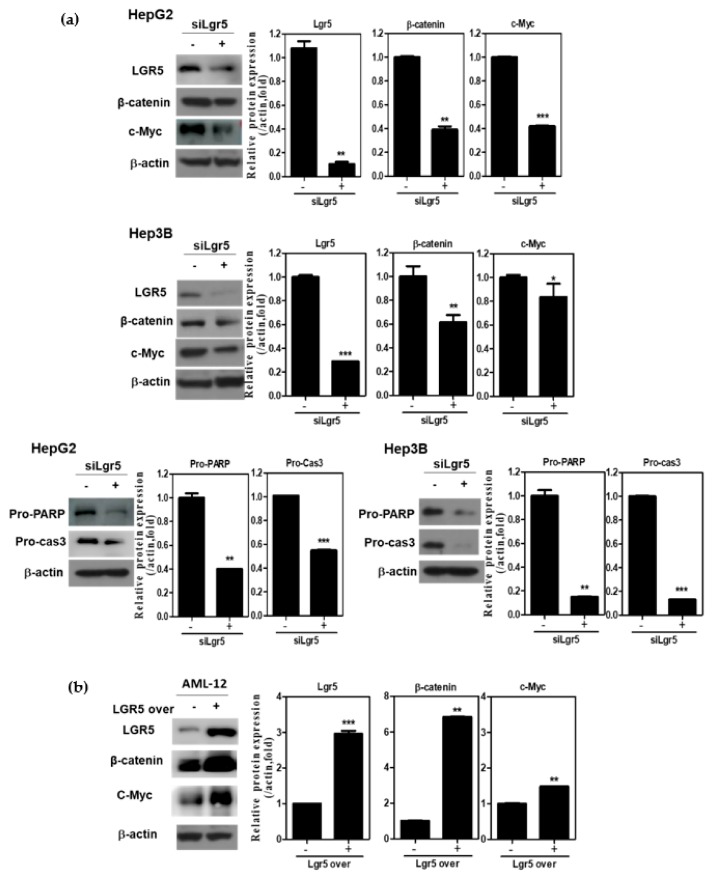
LGR5 depletion induces anti-proliferative and apoptotic effects in HCCs. (**a**) Effect of LGR5 depletion on LGR5 related proteins and apoptotic genes in HepG2 and Hep3B cells by Western blotting. Cell lysates were prepared and subjected to Western blotting for LGR5, β-atenin, c-Myc, PARP and Caspase3. (**b**) Effect of LGR5 overexpression on LGR5 related proteins in AML12 cells by Western blotting. AML-12 cells were transfected with LGR5 or control vector for 48 h. Cell lysates were prepared and subjected to Western blotting for LGR5, β-catenin and c-Myc. Graphs represent relative level of protein expression β-actin. Data represent means ± SD by two independent experiments. * *p* < 0.05, ** *p* < 0.01 and *** *p* < 0.001 vs. siRNA control.

**Figure 6 biomolecules-09-00718-f006:**
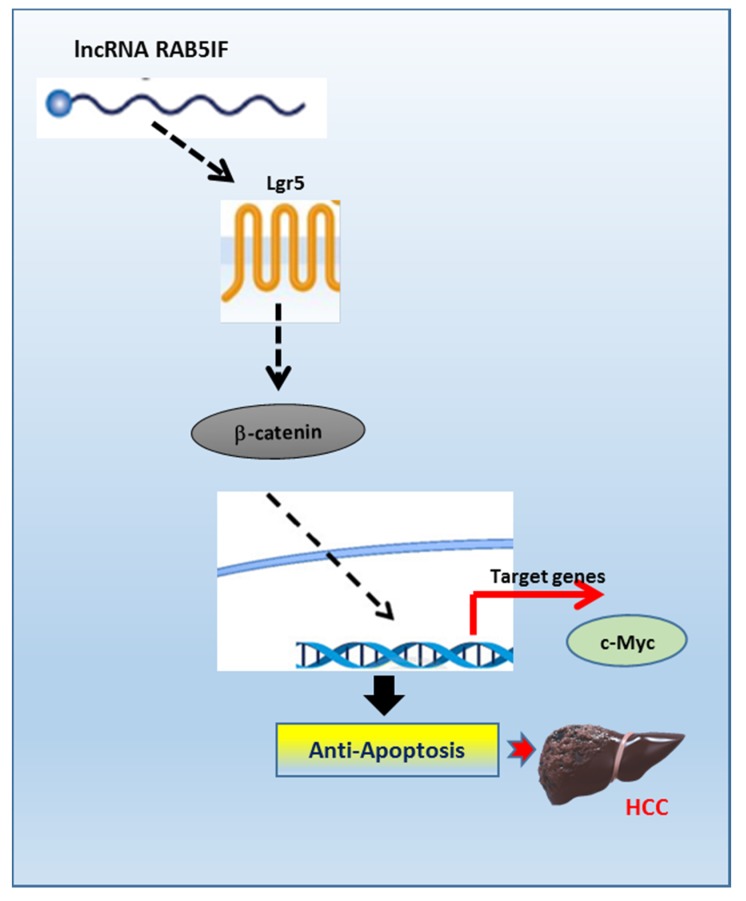
The pivotal role of LncRNA RAB5IF in the carcinogenesis of hepatocellular carcinoma. LncRNA RAB5IF promotes the growth of hepatocellular carcinoma cells via LGR5 mediated β-catenin and c-Myc signaling as a potent oncogenic target.

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
