# Peer review of "The Pivotal Role of Long Noncoding RNA RAB5IF in the Proliferation of Hepatocellular Carcinoma via LGR5 Mediated β-Catenin and c-Myc Signaling"

_biomolecules, 2019, doi:10.3390/biom9110718_

Round 1
Reviewer 1 Report
In this revised manuscript, Koo and colleagues did some efforts to address my comments; however, my feeling is that this research still lacks a clear rationale, and methods are not properly described.
To give an example, the information provided to describe the TGCA analysis are not sufficient to evaluate the appropriateness of the tools used and the soundness of the results shown. Was lncRNA RAB5IF among the top differentially expressed genes? Even if the authors prefer not to show a complete list of the genes found, they should give some comments to this point. To have an example from a paper tackling the same biological question, and that shows an appropriate description of the bioinformatic analysis performed the authors should read the paper of Falcon et al (2018) doi: 10.1155/2018/2864120.
Still there is confusion in indicating RAB5IF-protein or the corresponding lncRNA. See for example the title of paragraph 3.1, and in the Discussion, line 214.
In the abstract, the authors wrote: “Since long noncoding RNA (LncRNA) RAB5 interacting factor (RAB5IF) as a membrane protein was upregulated with close relationship with leucine-rich repeat-containing G protein-coupled receptor 5 (LGR5) as a same membrane protein and stem cell marker in hepatocellular carcinomas (HCCs) … " It is not very clear to me what the authors mean. Do RAB5IF and LGR5 belong to the same gene family? I could not find in the literature any evidence supporting this information. The authors should explain better the association between RAB5IF and LGR5. Regarding the same sentence, there is also a contradictory statement, as the authors say: “…long noncoding RNA (LncRNA) RAB5 interacting factor (RAB5IF)…” and then “..as a membrane protein”.
Finally, in the revised discussion, the authors hypothesize that lncRNA RAB5IF may act as a miRNA sponge for LGR5-targeting miRNAs. I understand that it would be challenging to demonstrate this hypothesis with experiments, but the authors should at least provide a preliminary in silico analysis to show that the lncRNA RAB5IF sequence possess some miRNA binding sites.
Minor:
On line 45: LncRNA RAB5IF (Accession number: NR_026562; Gene 46 symbol: C20orf24; Relationship: Exon sense-overlapping) at human chromosome 20. Although the sentence is intelligible, it lacks a verb.
Many typo errors are still present through the text.
Author Response
Comments and Suggestions for Authors
In this revised manuscript, Koo and colleagues did some efforts to address my comments; however, my feeling is that this research still lacks a clear rationale, and methods are not properly described.
To give an example, the information provided to describe the TGCA analysis are not sufficient to evaluate the appropriateness of the tools used and the soundness of the results shown. To have an example from a paper tackling the same biological question, and that shows an appropriate description of the bioinformatic analysis performed the authors should read the paper of Falcon et al (2018) doi: 10.1155/2018/2864120.
(Response) Thanks for your valuable comment. Bioinformatic analysis methods were updated based on Falcon’s paper you recommended.
Was lncRNA RAB5IF among the top differentially expressed genes? Even if the authors prefer not to show a complete list of the genes found, they should give some comments to this point.
(Response) Thanks. According to our previous LncRNA micoarray data only in HepG2 cells, LncRNAs such as HSP90B2P (Heat Shock Protein 90 Beta Family Member 2, Pseudogene), NACAP1(NACA Family Member 4, Pseudogene), ABCA11P (ATP Binding Cassette Subfamily A Member 11, Pseudogene), AURAPS1 (Aurora Kinase A Pseudogene 1), PTTG3P (Pituitary Tumor-Transforming 3, Pseudogene), RPL13AP17 (Ribosomal Protein L13a Pseudogene 17) and RAB5IF were upregulated in order in HepG2 cells. However, the other upregulated genes except LncRNA RAB5IF were found pseudogenes, while C20orf24, a gene symbol of LncRNA RAB5IF, was known to be involved in the progression of colorectal carcinoma cells as a membrane protein. Thus, the novel functions of LncRNA RAB5IF were examined in HCCs.
Still there is confusion in indicating RAB5IF-protein or the corresponding lncRNA. See for example the title of paragraph 3.1, and in the Discussion, line 214.
(Response) Sorry for making you confused. It was corrected as LncRNA RAB5IF in manuscript.
In the abstract, the authors wrote: “Since long noncoding RNA (LncRNA) RAB5 interacting factor (RAB5IF) as a membrane protein was upregulated with close relationship with leucine-rich repeat-containing G protein-coupled receptor 5 (LGR5) as a same membrane protein and stem cell marker in hepatocellular carcinomas (HCCs) … " It is not very clear to me what the authors mean. Do RAB5IF and LGR5 belong to the same gene family? I could not find in the literature any evidence supporting this information. The authors should explain better the association between RAB5IF and LGR5. Regarding the same sentence, there is also a contradictory statement, as the authors say: “…long noncoding RNA (LncRNA) RAB5 interacting factor (RAB5IF)” and then “..as a membrane protein”. Finally, in the revised discussion, the authors hypothesize that lncRNA RAB5IF may act as a miRNA sponge for LGR5-targeting miRNAs. I understand that it would be challenging to demonstrate this hypothesis with experiments, but the authors should at least provide a preliminary in silico analysis to show that the lncRNA RAB5IF sequence possess some miRNA binding sites.
(Response) Thanks for your critical comments. Frankly speaking, we were also confused about clear association mechanisms between LncRNA RAB5IF and LGR5, though depletion of LncRNA RAB5IF regulates LGR5 protein in HCCs in our study and also LncRNA RAB5IF was upregulated to 14.9 compared to average expression level (6.675) in HepG2 cells along with mRNA microarray data showing upregulation of LGR5 and beta catenin in HepG2 cells (data not shown). Despite our searching whether or not LncRNA RAB5IF sequence possesses some miRNA binding sites, we could not confirm the clear association between lncRNA RAB5IF and LGR5 protein due to lacking evidences. Thus, further study is requested to confirm the clear association mechanism between lncRNA RAB5IF and LGR5 protein using in silico analysis in the near future.
Minor:
On line 45: LncRNA RAB5IF (Accession number: NR_026562; Gene 46 symbol: C20orf24; Relationship: Exon sense-overlapping) at human chromosome 20. Although the sentence is intelligible, it lacks a verb.
(Response) Thanks. Corrected.
Many typo errors are still present through the text.
(Response) Thanks. Carefully corrected.
Reviewer 2 Report
The authors Koo et al. re-submitted the manuscript ID: biomolecules-622889 “The pivotal role of long noncoding RNA RAB5IF in the proliferation of hepatocellular carcinoma via LGR5 mediated β-catenin and c-Myc signaling”. The new version of the manuscript has been improved and it could be of helpful for the readers to understand the novel biological role of LcncRNA RAB5IF in the hepatocellular carcinoma by its interaction with LGR5, a leucine-rich repeat-containing G protein coupled receptor, stimulating β-catenin and c-Myc signaling.
In conclusion, I believe that the paper is now suitable for publication in biomolecules as it is.
Author Response
Comments and Suggestions for Authors
The authors Koo et al. re-submitted the manuscript ID: biomolecules-622889 “The pivotal role of long noncoding RNA RAB5IF in the proliferation of hepatocellular carcinoma via LGR5 mediated β-catenin and c-Myc signaling”. The new version of the manuscript has been improved and it could be of helpful for the readers to understand the novel biological role of LcncRNA RAB5IF in the hepatocellular carcinoma by its interaction with LGR5, a leucine-rich repeat-containing G protein coupled receptor, stimulating β-catenin and c-Myc signaling.
In conclusion, I believe that the paper is now suitable for publication in biomolecules as it is.
(Response) Thanks a lot for your positive comment.
Round 2
Reviewer 1 Report
The authors addressed my comments, I therefore propose this manuscript for publication on Biomolecules
This manuscript is a resubmission of an earlier submission. The following is a list of the peer review reports and author responses from that submission.
Round 1
Reviewer 1 Report
The authors Ja Il Koo et al., has been submitted a manuscript ID: molecules-600193 “The pivotal role of long noncoding RNA RAB5IF in the proliferation of hepatocellular carcinoma via LGR5 mediated β-catenin and c-Myc signaling” This manuscript could help to understand the different biological role of LcncRNA (RAB5) in hepatocellular carcinoma by interaction with LGR5 a leucine-rich repeat-containing G protein coupled receptor stimolating β-catenin and c-Myc signaling.
Major points to be improved.
In the NCBI databank, RAB5 interacting factor (RAB5IF) gene is reported as a protein coding gene, although some transcript variants are present as non-coding RNAs likely involved in Non-sense Mediated Decay. In this regard, the authors should clarify why they report RAB5IF as a long non-coding RNA and they should report the accession number of the specific transcript they analyzed in the materials and method section. In the Fig 1a, the tissue tumor or control analyzed are not described. The authors should provide a better description. The materials and methods section need to be strongly improved including many missing details and references. For the qPCR analysis of the LGR5 gene expression, the primers are not reported.
In conclusion, I believe that the paper need extensive changes to be suitable for publication.
Reviewer 2 Report
In this manuscript, Koo, Lee and colleagues describe a role for the long non-coding RNA (lncRNA) RAB5IF in the progression of hepatocellular carcinoma (HCC) via LGR5-mediated b-catenin and c-Myc pathways. A bioinformatic analysis of the The Cancer Genome Atlas (TCGA) revealed that the lncRNA RAB5IF is overexpressed in HCC and this correlates with a poor survival rate. RAB5IF depletion reduced the expression of LGR5 and its downstreams such as b-catenin, c-Myc in HepG2 and Hep3B cells. Notably, LGR5 depletion also attenuated the expression of pro-PARP, pro-caspase3, b-catenin and c-Myc in HepG2 and Hep3B cells. Conversely, LGR5 overexpression upregulated b-catenin and c-Myc in AML-12 normal hepatocytes. These finding led the authors to conclude that LncRNA RAB5IF promotes the growth hepatocellular carcinoma cells via LGR5 mediated b-catenin and c-Myc signaling as a potent oncogenic target.
As a general comment, I am not sure this manuscript falls within the topics covered by the Molecules Journal. The MDPI Non-Coding RNA journal would certainly be more appropriate and would meet a wider audience.
My main criticism is that the rationale of this research is not very clear.
The authors start their research by interrogating The Cancer Genome Atlas (TCGA). In their analysis they found that expression of the long noncoding RNA RAB5IF is increased in HCC and that the high expression correlates with poor survival (Fig. 1). However, the bioinformatic analysis that led to this discovery is not described in the Methods section. Second, no information on the long non-coding RNA RAB5IF gene is provided. Does it have an official name or symbol? Is it classified as a “natural antisense” (NAT) lncRNA?
Is there any sequence deposited on GeneBank? This would be important, for example, to understand how the siRNA targeting RAB5IF lncRNA were designed.
The use of “RAB5IF” to indicate both the protein-coding gene and the associated lncRNA is confusing. For example, in fig. 2 and 3 the authors performed siRNA experiments targeting the RAB5IF lncRNA, but the legends report siRAB5IF, which led to assume that experiments where performed by silencing the protein-coding gene.
Does the downregulation of RAB5IF lncRNA affects the expression levels of RAB5IF protein? This would be very important to support the authors’ hypothesis.
Otherwise it is not clear how the lncRNA RAB5IF can impact the function of lgr5 (according the cartoon showed in Figure 6)
Minor comment:
The introduction is too succinct, and it does not allow to fully understand the rationale of the research.